# On the Effects of Data Distortion on Model Analysis and Training

## Abstract

Data modification can introduce artificial information. It is often assumed that the resulting artefacts are detrimental to training, whilst being negligible when analysing models. We investigate these assumptions and conclude that in some cases they are unfounded and lead to incorrect results. Specifically, we show current shape bias identification methods and occlusion robustness measures are biased and propose a fairer alternative for the latter. Subsequently, through a series of experiments we seek to correct and strengthen the community's perception of how distorting data affects learning. Based on our empirical results we argue that the impact of the artefacts must be understood and exploited rather than eliminated.

## 1 Motivation

Modifying data has become commonplace both when training and analysing models, yet the wider implications are often disregarded. We delve into some of the side-effects and point out that this practice has resulted in the creation of biased model interpretation tools and poorly informed theories. On the analysis side, we take as examples occlusion robustness and shape bias identification methods. On the training side, we focus on some instances of Mixed Sample Data Augmentation (MSDA), where two images are combined to obtain a new training sample. Visual examples of each can be found in Figure 1. In this paper we study a number of assumptions that lie at the heart of the aforementioned methods, which we briefly introduce below.

**Shape-texture bias:** Deep models are known to be sensitive to interventions that are imperceptible to humans [35, 13], as well as to other forms of distribution shifts [1, 6, 8]. It has been argued that this is intimately linked to networks tending to use texture rather than shape information [2, 11]. Recently, input distortions have become a popular way of assessing a model's texture bias. To this end, images are divided into a grid and the resulting patches are randomly shuffled such that information is preserved locally, while the global shape is altered [32, 27, 25, 41]. It is implicitly assumed that patch-shuffling does not introduce misleading shape or texture that could affect model evaluation.

**Occlusion robustness:** A widely adopted method for measuring occlusion robustness is through the accuracy obtained after superimposing a rectangular patch on an image [5, 9, 39, 42, 20]. We refer to this approach as CutOcclusion throughout the paper. Just as for shape bias, this method relies on information introduced not to interfere with a model's learnt representations such that a decrease in performance can be directly attributed to lack of robustness.

**Data augmentation studies:** In statistical learning, training with augmented data is termed Vicinal Risk Minimisation (VRM) [37, 4] and it is seen as injecting prior knowledge about the neighbourhood of the data samples. The intuition behind augmentation caused researchers to interpret its effect through the similarity between original and augmented data distributions. This perspective is often challenged by methods which, despite generating samples that do not appear to fall under the

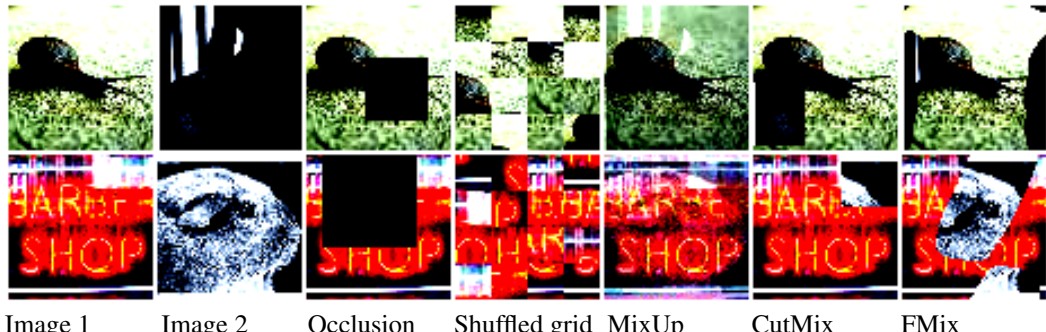

| Image 1 | Image 2 | Occlusion | Shuffled grid | MixUp | CutMix | FMix |

Figure 1: Examples of image distortions. For test-time distortions (Occlusion and Shuffled grid) only Image 1 was used. For mixing augmentations, the first row was generated with a mixing factor of 0.2, while the second one with 0.5.

distribution of natural images, lead to strong learners. Gontijo-Lopes et al. [12] argue it is the *perceived* distribution shift that needs to be minimised, while maximising the sample vicinity. Formalising these concepts, they introduce augmentation "diversity" and "affinity". Diversity is defined as the training loss when learning with artificial samples, while affinity quantifies the difference between the accuracy on original test data and augmented test data for a reference model. The latter penalises augmentations that introduce artificial information to which the model is not invariant, implicitly assuming that training with that information is detrimental to generalisation.

In summary, it is currently assumed that the artefacts introduced by changes in the data are negligible when evaluating models, while those introduced when training are important and undesirable. Does the artificial information added by analysis methods not have major side-effects or does it lead to biased results? Conversely, are the artefacts important when training with modified data? Do they cause models to learn better or worse representations?

We set out to answer these questions and find that when the secondary effects of data manipulation are not accounted for, results can be misleading, especially in comparative studies. Subsequently, we construct empirical counter-examples which disprove common beliefs in the literature and highlight the importance of understanding the changes MSDAs introduce. Our contributions are:

- We show that increasingly popular model interpretation and analysis methods are biased, relying on unfounded assumptions (Section 2);

- For measuring occlusion robustness, we propose a fairer alternative (Section 3);

- We show that, in contrast to what is widely assumed, not preserving the data distribution can lead to learning better representations (Section 4).

## 2   Are artefacts negligible when analysing classifiers?

To verify whether distorting data at evaluation time could have side-effects previously not considered, we look at the increase in misclassifications per category. That is, from the number of incorrect predictions of a model evaluated on modified data, we subtract the incorrect predictions when testing on original data. If there is a significant increase for a specific class, it indicates that the distortion introduces features the model associates with that class. We refer to this phenomenon as "data interference". Considering only positive differences, we denote the increase in the percentage of misclassifications for class $c$ of a given model $m$ by $i_c^m$. We define the Data Interference $DI$ index as

$$DI = \frac{i_{c_{max}}}{\sum_c i_c} i_{c_{max}}, \tag{1}$$

where $c_{max}$ is the class with highest mean increase across all runs. The $DI$ index measures the percentage represented by the dominant class $c_{max}$ weighted by its increase. A high index value indicates a sharp increase for a particular class which is consistent across runs. We associate this with an overlap between introduced artefacts and learnt representations. In Appendix B.1 we experiment as

Table 1: DI index (%) for PreAct-ResNet18 on grid-shuffled images for four different types of models. Results with the highest average are given in italic and the lowest in bold. Information introduced when shuffling tends to interfere less with the representations of FMix and CutMix models.

|  | basic | MixUp | FMix | CutMix |
|---|---|---|---|---|
| CIFAR-10 | $2.90_{\pm 1.10}$ | $2.54_{\pm 1.29}$ | $0.60_{\pm 0.26}$ | $\mathbf{0.33_{\pm 0.17}}$ |
| CIFAR-100 | $1.05_{\pm 0.59}$ | $0.93_{\pm 0.44}$ | $0.23_{\pm 0.29}$ | $\mathbf{0.11_{\pm 0.10}}$ |
| FashionMNIST | $1.12_{\pm 0.63}$ | $2.73_{\pm 1.64}$ | $1.12_{\pm 0.61}$ | $\mathbf{0.70_{\pm 0.22}}$ |
| Tiny ImageNet | $2.58_{\pm 4.73}$ | $0.54_{\pm 0.27}$ | $0.38_{\pm 0.12}$ | $\mathbf{0.14_{\pm 0.12}}$ |
| ImageNet | $0.82$ | $1.49$ | $\mathbf{0.58}$ | — |

with an alternative index, where we weight by the highest increase of a model across the 5 runs, so as to obtain a worst-case analysis. As expected, we observe a more accentuated bias in this case.

To obtain models with different behaviours in a controlled manner, we make use of data augmentation. Since it is sufficient to identify some common cases in which models are disfavoured, we choose to reduce our environmental impact by restricting the analysis to simple MSDAs that combine images without incurring additional computation time or external models. As will be argued in Section 3, we expect the unfairness to be present in most settings, thus the exact choice of augmentation is irrelevant. We focus on two popular MSDAs, MixUp [40] and CutMix [39]. MixUp linearly interpolates between two images to obtain a new training example, while CutMix masks out a rectangular region of an image with the corresponding region of another image. Besides the aforementioned methods, we also employ FMix due to its irregularly shaped masks sampled from Fourier space, which will play an important role in our analysis. Note that although the masking methods sample the size of the occluding patch from the same distribution, in CutMix part of the rectangle can be outside the image, which leads to less occluded samples overall. We refer to models by the augmentations they were trained with and use "basic" to label the models trained without MSDA.

Throughout the paper, we do 5 runs of each experiment with PreAct-ResNet18 [15] as the default architecture. We include results for BagNet [2] and VGG [33] in the Appendices. The main data sets we report results on are CIFAR-10/100 [21], Tiny ImageNet [34], FashionMNIST [38], ImageNet [29]. For ImageNet we use pretrained ResNet-101 models made publicly available by Harris et al. [14]. Note that the only experiments for which we are unable to run repeats are those on ImageNet, since only one model per augmentation is provided. For full experimental details, see Appendix A. Total emissions of training the models evaluated in this paper are estimated [22] to be 38.07 kgCO$_2$eq, to which 13.11 kgCO$_2$eq more are added during the analysis. The hope is that our findings will lead to a better understanding which in the long run would help reduce erroneous research directions without needing to empirically disprove them, lessening the future carbon footprint of the community.

### 2.1 Shape bias

For assessing shape bias through sample manipulation, the standard procedure is to choose between dividing the image in 4, 16 or 64 patches to be shuffled. Since FashionMNIST images are smaller, we choose a $2 \times 2$ grid, for CIFAR-10/100 and Tiny ImageNet $4 \times 4$, while for ImageNet we use an $8 \times 8$ grid. However, similar results are obtained for different grid sizes (Appendix B.2). The large DI index in Table 1 indicates that either basic or MixUp models tend to associate the features artificially introduced by patch-shuffling with a certain class. We take a closer look at the distribution of misclassifications for CIFAR-10 and notice that the basic model tends to wrongly predict the class "Truck" (Figure 2). This is not at all surprising, given that the strong horizontal and vertical edges are highly indicative of this class. Similar observations can be made for other data sets (Appendix B.3). Thus, we believe the grid-shuffling approach is causing models which are not invariant to strong horizontal and vertical edges to appear to rely more heavily on shape information. A model not affected by this transformation could be considered texture-biased if we accept the larger definition of texture as local information. However, there is a question about the extent to which the reciprocal is true; A model can be invariant to the aforementioned edges because it is indeed relying on texture information or simply because it uses different shape-related features.

**Is a model necessarily more affected by patch-shuffling if it has a higher shape bias?** To answer this question, we can use another method of determining shape and texture bias to find a counter-

Table 2: Accuracy of augmentation-trained ImageNet and Tiny ImageNet models on the GST data set when the label is taken to be either the shape or texture. There is no clear correlation between masking methods and low texture bias.

| | ImageNet | | Tiny ImageNet | |
|---|---|---|---|---|
| | Shape | Texture | Shape | Texture |
| basic | 20.31 | 53.28 | $10.56_{\pm 0.65}$ | $26.04_{\pm 1.77}$ |
| MixUp | 24.14 | 60.31 | $12.02_{\pm 0.33}$ | $27.77_{\pm 1.56}$ |
| FMix | 21.25 | 53.43 | $10.40_{\pm 0.39}$ | $19.90_{\pm 2.12}$ |
| CutMix | — | — | $10.54_{\pm 0.38}$ | $23.72_{\pm 2.42}$ |

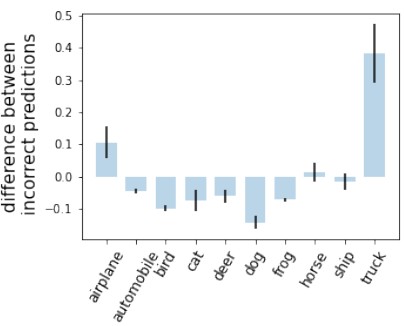

Figure 2: Difference in incorrect predictions on CIFAR-10 for the basic model.

example. We analyse the ImageNet models on the Geirhos Style-Transfer (GST) [11] data set. GST contains artificially generated images where the shape belongs to one class and the texture to another. There are 16 coarse classes that encompass a number of ImageNet categories to which they are mapped. The bias of the models is given by the accuracy obtained when the label is set to either the shape or texture. Using this well-known method of identifying shape bias we want to find models which have similar biases but different DI indices when patch-shuffling. This would indicate that sensitivity to shuffling is not necessarily linked to increased shape bias.

The results in Table 2 show that the basic model does not have a higher shape bias than masking methods although it has a significantly higher DI index, as we have seen in Table 1. We repeat the same experiment on the Tiny ImageNet [34] data set. Geirhos et al. [11] use WordNet [26] to map the 1000 categories to the 16 classes of the GST data set. A number of ImageNet categories that belong to the 16 higher-level classes of GST are missing. For this reason, a poorer overall performance is expected and the results could differ slightly given a better fit between the sets. Nonetheless, we find again no significant correlation between masking augmentation and texture bias. We also include in Table B.2 the results for BagNet models, which have smaller receptive fields and so are forced to use more local information. Even in this case, we find a high DI for the basic model and no difference in texture bias compared to MSDA. Thus, a model which is more affected by patch-shuffling is not necessary more shaped-bias. In other words, models can appear to have vastly different shape bias when evaluated on randomly rearranged patches, albeit in reality their bias is similar. The inability of this method to account for artefacts makes it unfair and unreliable.

## 2.2 Occlusion measurement

We want to determine whether the same issue identified in the case of shape bias evaluation applies to occlusion robustness measures. We focus on CutOcclusion, where a rectangular black patch is superimposed on test images and the robustness is given by the resulting accuracy. We perform the same experiment as before, where the $DI$ index is now measured when testing on rectangle-occluded images. There is no standardised distortion when measuring CutOcclusion, with the size and positioning of the obstructing patch varying between studies. Most often in prior art a lack of robustness is noted for large occluders [e.g. 5, 42]. For this reason, we sample the size of the patch from a Beta(2,1) distribution, allowing the occluding patch to lie outside the image (as it is done for augmenting with CutMix and CutOut [7]). This allows us to capture both the cases in which either the centre or the border area is masked out but requires a non-uniform distribution to counter for the patches existing outside the image. We also sample from a uniform distribution where the occluder is restricted to be positioned within the image boundaries and obtain similar results (See Appendix C.1).

Table 3 shows that a significant gap in the DI index can be identified for each of the data sets. This indicates that some models will again be disadvantaged. We additionally find that data interference is present for different architectures, when overlapping patches from external images or using differently shaped masks (Appendix B.4). Thus, the result of CutOcclusion and its variants is highly dependent on the problem at hand. That is, whether the artefacts introduced by the artificial occlusion are salient features of the model depends on what features are naturally distinctive for the model. Just as for randomly shuffling tiles, by occluding images using a particularly shaped patch, one implicitly

Table 3: DI index (%) when occluding with black patches. The highest results are given in italic and the lowest in bold. For each data set, there exists a non-negligible gap in the DI index.

|  | basic | MixUp | FMix | CutMix |
|---|---|---|---|---|
| CIFAR-10 | $5.48_{\pm 1.31}$ | $\mathbf{0.75_{\pm 0.55}}$ | $0.87_{\pm 0.92}$ | $2.87_{\pm 3.04}$ |
| CIFAR-100 | $3.25_{\pm 1.31}$ | $\mathbf{0.80_{\pm 0.46}}$ | $1.28_{\pm 2.48}$ | $1.36_{\pm 0.63}$ |
| FashionMNIST | $\mathbf{0.44_{\pm 0.34}}$ | $2.59_{\pm 1.05}$ | $0.92_{\pm 1.79}$ | $0.97_{\pm 1.82}$ |
| Tiny | $2.40_{\pm 0.82}$ | $1.88_{\pm 0.61}$ | $\mathbf{0.47_{\pm 0.41}}$ | $4.62_{\pm 4.93}$ |
| Imagenet | $1.28$ | $4.50$ | $\mathbf{1.02}$ | — |

measures a model's affinity to certain features, albeit those features might be discriminative. This deems such methods inappropriate for fairly assessing robustness and texture bias.

A related observation was made by Hooker et al. [17] who note the pitfalls of manipulating data to determine feature importance. They point out that when simply superimposing uniform patches over image features, it is difficult to asses how much of the reduction in accuracy is caused by the absence of those features and how much is due to images becoming out of distribution. To address this, the most important features identified by an estimator are masked out both on train and test data, closing the gap between the two sets. Hooker et al. then train and evaluate models on the newly generated images. Unlike for interpretability methods, the subject of occlusion robustness studies is the model itself, which makes training with a modified version of the data an inviable option. In the following section we explore ways of overcoming this bias when measuring occlusion robustness.

## 3    What are fairer alternatives?

We propose a simple, more carefully defined measure that aims to decouple the machine's edge bias from the occlusion robustness, which we refer to as "interplay occlusion" (iOcclusion). Interplay occlusion reflects the change in the interplay between performance on seen and unseen data. Formally,

$$iOcclusion_i = \left| \frac{\mathcal{A}(\mathcal{D}^i_{train}) - \mathcal{A}(\mathcal{D}^i_{test})}{\mathcal{A}(\mathcal{D}_{train}) - \mathcal{A}(\mathcal{D}_{test})} \right|, \tag{2}$$

where $\mathcal{A}(\mathcal{D})$ denotes the accuracy on a given data set $\mathcal{D}$, and $\mathcal{D}^i$ is the data set resulting from removing $i\%$ pixels of each image. The intuition is that on train data robust models are less sensitive to the artefacts of the occlusion policy for small levels of occlusion, resulting in a large difference in accuracy from that on unseen data. The performance of both train and test gets close to random as the percentage of occluded data approaches 90% and we expect the gap to fall off quicker for less robust models. This change in interplay is taken with respect to the generalisation gap of the model, such that the quality of the model fit in itself does not interfere with the robustness measure.

Although iOcclusion reduces data interference, other factors have to also be considered when choosing a masking method for computing $\mathcal{D}^i$, such as the number of contiguous components or the amount of salient information masked out. In this paper we choose to generate masks using Grad-CAM [30], such that the area with most salient i% pixels is covered. It must be noted that this method implicitly assumes there could be multiple occluders and has the downside of incurring a higher environmental cost. For this, we also experiment with using rectangular or Fourier-sampled masks and conclude that although random masking makes the process noisier, the exact choice of masking method is of secondary importance as long as the occluder's granularity is accounted for. Appendix C.4 provides discussion and results on these alternative instances of iOcclusion, as well as differences in their carbon footprint. For a fair comparison, throughout this section we do not allow the obstructing patch when measuring CutOcclusion to lie outside the image such that the fraction removed is exact.

Assessing the correctness of such a measure is difficult in the absence of a baseline. For the remainder of this section we will build varied experiments to attest the validity of our method. We focus on the key results, but include additional ones and further experiments in Appendix C. Since occlusion in real-life scenarios could be caused by non-uniformly coloured objects, an appropriate measure must generalise across colour patterns. When computing iOcclusion and CutOcclusion, we superimpose patches from images belonging to a different data set and compare the results to those obtained when occluding with black patches only. For visual clarity, Figure 3 presents the results for the

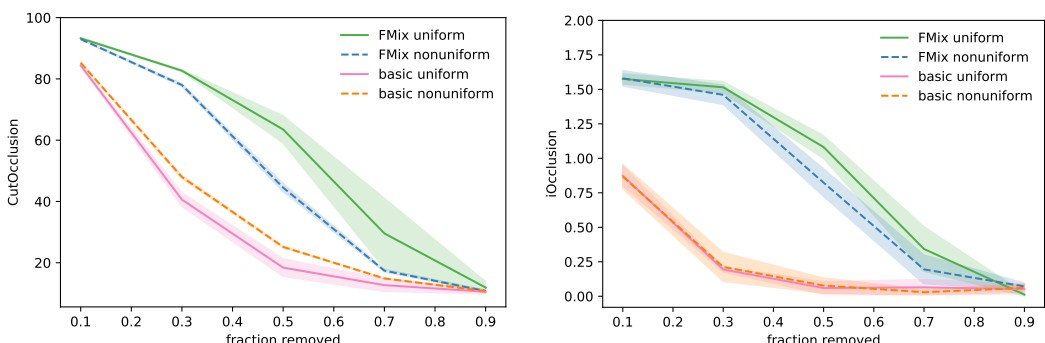

Figure 3: CutOcclusion (left) and iOcclusion (right) when occluding with black patches (uniform) and patches taken from other images (nonuniform). iOcclusion gives more consistent results.

Table 4: DI index (%) and occlusion robustness for models trained on CIFAR-10 when obstructing 30% of the image pixels with non-uniform patches. When measuring the robustness with CutOcclusion, RM appears significantly less robust than CutMix due to its sensitivity to patching with rectangles, while iOcclusion highlights the robustness specific to training with FMix-like masks. Given in bold is the closest result to that of RM for each evaluation.

|  | basic | MixUp | CutMix | FMix | RM |
|---|---|---|---|---|---|
| DI index | $1.67_{\pm 0.27}$ | $1.00_{\pm 0.31}$ | $0.17_{\pm 0.16}$ | $0.15_{\pm 0.03}$ | $0.39_{\pm 0.07}$ |
| CutOcclusion | $47.97_{\pm 0.52}$ | $\mathbf{58.65_{\pm 1.01}}$ | $76.56_{\pm 6.36}$ | $78.00_{\pm 0.45}$ | $60.79_{\pm 5.03}$ |
| iOcclusion | $0.21_{\pm 0.10}$ | $0.57_{\pm 0.18}$ | $\mathbf{1.09_{\pm 0.17}}$ | $1.46_{\pm 0.07}$ | $1.20_{\pm 0.23}$ |

least and most robust models (see Appendix C.2 for a full comparison). For iOcclusion, using uniform occluders gives similar results to its non-uniform version, whereas the CutOcclusion measure provides an inconsistent model evaluation.

As we have argued, in addition to not being sensitive to the colour pattern of the patch, a fair measure must also be invariant to the shape of the patch. To empirically confirm iOcclusion reduces the importance of edge information, we aim to obtain a model that is robust to occlusion, but at the same time has a high DI index (it is sensitive to edge information). To this end, we create a variation of FMix, Random Masks (RM), where at the beginning of the training process three masks are randomly sampled from Fourier space. For each batch, one of the three is chosen uniformly at random. While the RM models are not sensitive to black-patch occlusion, when masking with patterned patches they have a higher DI index than FMix, as desired. Table 4 gives results for a fraction of 0.3 pixels covered by a non-uniform occluder. Our measure reflects the robustness of training with RM, situating it close to other masking methods. On the other hand, because CutOcclusion implicitly penalises models with high DI index, according to this measure RM appears almost as sensitive to occlusion as MixUp. Figure B.4 shows results for a wider range of fractions.

Another problem that occurs when purely looking at post-masking accuracy is weaker models would erroneously appear less robust. We show this by reversing the problem: we evaluate the same model on two different subsets of the CIFAR-100 data set: typical and tail images as categorised by Feldman and Zhang [10]. They consider a train-test sample pair to belong to the tail of the data distribution if the test sample is correctly classified when a model is trained with the train sample and incorrectly without it. CutOcclusion would indicate that models are significantly more robust to occluding typical examples. However, a closer analysis makes us doubt this conclusion. The raw accuracy on both train and test data for tail examples is lower than for the typical ones. In fact, the performance when occluding images decreases at the same rate for the two subsets. By way of definition, iOcclusion allows a fair comparison of robustness regardless of the overall performance of a model (Figure 4).

As we evidenced through controlled experiments, there are many cases that CutOcclusion does not properly address. From a model analysis perspective, correctly assessing the occlusion robustness could lead to better understanding and development of models and training procedures. Equally important, it has applicability for real-world deployments where no prior knowledge exists about the

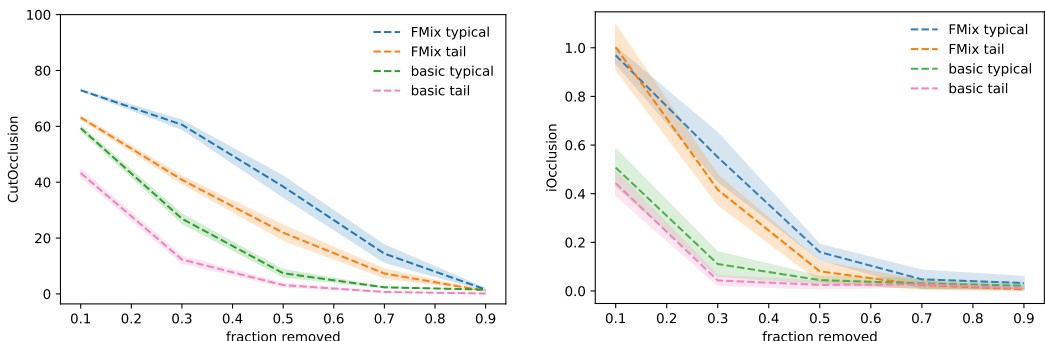

Figure 4: CutOcclusion (left) and iOcclusion (right) for the basic and FMix models on two subsets of the same data set: tail and typical. Evaluating the models with iOcclusion on the two types of samples leads to mostly overlapping robustness levels. That is, they do not differ outside the margin of error. On the contrary, CutOcclusion incorrectly finds the models to be less robust on tail data.

Table 5: Augmentation comparison on CIFAR-10. We consider two variants when calculating diversity. One is computing the cross-entropy loss using the label of the majority class (Diversity), as for mixing in [19]. The alternative, MixDiversity, takes a linear combination of the two cross-entropy losses.

Table 6: DI index (%) measured for non-uniform occlusion when training without the class with highest increase in incorrect predictions. Again, a gap can be noted, supporting the idea that data interference is not specific to peculiar cases.

| | Affinity | Diversity | MixDiversity | | CIFAR-10 | CIFAR-100 |
|---|---|---|---|---|---|---|
| MixUp | $-12.58_{\pm 0.14}$ | $0.41_{\pm 0.01}$ | $0.84_{\pm 0.00}$ | MixUp | $0.39_{\pm 0.15}$ | $1.22_{\pm 0.19}$ |
| FMix | $-25.55_{\pm 0.26}$ | $0.34_{\pm 0.01}$ | $0.65_{\pm 0.00}$ | FMix | $0.08_{\pm 0.06}$ | $0.50_{\pm 0.21}$ |

possible shapes of the obstructions. While this aspect of generality is the strength of our approach, it must be stressed that when there exists a limited set of known possible occluders, evaluating robustness specifically to them could be safer. Incorrectly assessing robustness can have severe effects especially when applied to autonomous vehicles or medical imaging. We do not propose a universal solution, but rather suggest an alternative to the biased approach for the common scenario in which the environment is not controlled and little is known about all the potential occluders. However, even in this case, our metric should be taken as a guide when analysing models. Although iOcclusion aims to address data interference, since a ground-truth does not exist, it cannot be guaranteed that this method provides fair results in the absolute.

The strength of the bias will depend on the data in question and some applications will be more heavily affected than others. We have seen that for natural images this bias does exist. To confirm that we have not just identified isolated cases, we remove the class that has the highest increase in mispredictions and retrain the models on the remaining classes. We find that the bias is again present (Table 6), but with respect to another class. For example, in the case of CIFAR-10, after removing the "Truck" class, models mispredict rectangle-occluded images as "Boat" (Appendix D). Thus, the edge artefacts are very likely to interfere with learnt representations since they are such fundamental features. From an evaluation perspective, as we have seen, this impacts assessment methods and must be accounted for. From the training perspective, such a widespread data interference of masking distortions would indicate a large perceptual shift in the data when performing MSDA. In the following section we investigate the importance of the artefacts in this case and their implications.

## 4  Is the magnitude of the distribution shift important?

Traditionally it was believed that a good augmentation should have minimal distribution shift. Most recently, it has been argued that it is the degree of the perceived shift that determines augmentation quality [12]. We start with the perceptual gap of training with MSDA, as proposed in Gontijo-Lopes et al. [12]. Reiterating, this is given by the difference between the performance of the baseline model when presented with original test data and augmented test data and is termed "affinity". Subsequently,

we address the gap in the wider sense, as is often sought in prior art. We first argue that high affinity and high diversity are not necessarily desirable. Indeed, on CIFAR-10, we find FMix, a better performing augmentation, to have both lower affinity and lower diversity than MixUp (Table 5). For diversity, we compute the cross-entropy loss where the label is taken to be that of the majority class. Similar results are obtained with the MixUp loss, where a weighted average of the true labels is taken.

While intuitively for a high level of affinity, high diversity could correspond to better methods, the converse does not hold. We argue this is because affinity is rather an analysis of the learnt representations of the reference model and cannot give an insight into the quality of the augmentation or its effect on learning. The limitations of affinity are intimately linked to those of CutOcclusion. We have seen in Section 3 that the bias of the basic model is present not only when obstructing an image with a uniform patch, but also when mask-mixing. As such, an augmentation will have a lower affinity if it introduces artefacts that could otherwise lead to learning better representations when used in the training process. We believe this issue extends to other approaches that aim to motivate the success of MSDA through reduced distribution shift. Henceforth, we focus on bringing further supporting evidence that the importance lies in the invariance introduced by the shift and its interaction with the given problem rather than its magnitude.

### 4.1   If it is not the magnitude that matters, is it the direction?

We use empirical evidence to argue against previous assumptions behind the success of MSDA and propose the study of introduced bias as a more informative research direction. Here we use the term "bias" to refer to a drift in the learnt representations introduced by the change in the training procedure. A fundamental difference to classical training is that in the case of augmentation the samples are no longer independent. Mixed-sampling takes this even further. An immediate question is, does the added correlation lead to more meaningful representations? It is claimed that the strength of MixUp lies in causing the model to behave linearly between two images [40] or in pushing the examples towards their mean [3]. Both of these claims rely on the combined images to be generated from the same distribution. We want to verify to which extent this is necessary for a successful augmentation.

It has been argued that label mixing has a negligible effect on the final model performance [19, 18, 14, 24]. We use the reformulated objective setting [18, 14], where targets are not mixed and the mixing coefficient is drawn from an imbalanced Beta distribution. This allows us to apply MSDA between data sets. Thus, for training a model on a data set, we use an additional one whose targets will be ignored. As an example, a model that is learning to predict CIFAR-10 images will be trained on a combination of CIFAR-10 and CIFAR-100 images, with the target of the former. This scenario breaks the added correlation between training examples. Note that when mixing between data sets we use the same procedure as when performing regular MSDA, without improving the process.

Table 7 contains the results of this experiment, showing that an accuracy similar to or better than that of regular MSDA can be obtained by performing inter-dataset MSDA.This invalidates the argument that the power of MixUp resides in causing the model to act linearly between samples. Another observation is that for FMix and MixUp, introducing elements from CIFAR-100 when training models on the CIFAR-10 problem does not harm the learning process. The reciprocal, however, does not hold. Hence, the "distribution shift" is more intimately linked to the problem at hand and aiming to characterise an augmentation based on the distance from the original distribution is a limiting approach, especially when the distance is measured as perceived by a reference model.

We believe an explanation is that the artefacts created when putting together images from CIFAR-10 with those of CIFAR-100 could introduce information that makes the separation of the 10 classes easier. However, if the information happens to interfere with a feature that is important for separating the CIFAR-100 categories, the performance could degrade on this data set. This singular experiment is not sufficient to draw any general conclusions. However, it does show that shifting two distributions by the same amount can have different effects on the model performance. Thus, the specifics of the bias introduced could be more important than its magnitude. While some level of data similarity has to be preserved when performing MSDA, it is far from being the objective of such data-distorting approaches which, as we will argue further, should be rather seen as forms of regularisation.

We have seen that for all considered data sets, artefacts introduced by masking methods seem to overlap with common features. This has led us to believe that MSDA training could help bypass some of the simplicity bias. The simplicity bias refers to the tendency of deep models to find simple

Table 7: Accuracy on CIFAR-10 (left) and CIFAR-100 (right) upon mixing with samples from a different data set. The baseline is the accuracy when training with a single data set using the reformulated objective. In the interest of space, CIFAR-110 is used to refer to mixing with CIFAR-100 when training on the CIFAR-10 problem and vice-versa.

|  | MixUp | FMix | CutMix | MixUp | FMix | CutMix |
|---|---|---|---|---|---|---|
| baseline | $94.18_{\pm0.34}$ | $94.36_{\pm0.28}$ | $94.67_{\pm0.20}$ | $74.68_{\pm0.37}$ | $75.75_{\pm0.31}$ | $74.19_{\pm0.50}$ |
| CIFAR-110 | $94.70_{\pm0.27}$ | $94.80_{\pm0.32}$ | $94.66_{\pm0.12}$ | $72.36_{\pm1.04}$ | $74.80_{\pm0.55}$ | $74.47_{\pm0.39}$ |
| Fashion | $92.28_{\pm0.28}$ | $95.03_{\pm0.10}$ | $94.61_{\pm0.19}$ | $66.40_{\pm1.86}$ | $74.46_{\pm0.57}$ | $74.06_{\pm0.28}$ |

representations and has been used to justify the success of deep models [28, 36]. Recent research shows that this propensity causes models to ignore complex features that explain the data well in favour of elementary features, even when they lead to worse performance [31, 16].

Although it could seem natural that since MSDAs are not augmentations in the VRM sense, they will increase the complexity of the problem, we design an experiment to support this claim. Similarly to Shah et al. [31], we combine CIFAR-10 and MNIST [23] samples. Since they have the same number of classes, we can easily associate each class of one data set with a corresponding one from the other. Thus, we stack a padded image from the $k$th class of MNIST on top of a sample from the $k$th class of CIFAR-10, such that a $3 \times 64 \times 32$ image is obtained. We then randomly combine the test images and separately compute the accuracy with respect to the targets of each data set.

The predictions with respect to the CIFAR-10 labels are no better than random ($10.04_{\pm0.11}$), while the accuracy with respect to the MNIST images remains high ($99.57_{\pm0.72}$). Thus, models trained on this combination are mostly relying on MNIST images to make predictions. Similar behaviours have previously been associated with simplicity bias. Subsequently, when training, we perform FMix only on MNIST images and observe that this is enough to reverse the results. Evaluating against the CIFAR-10 label gives an accuracy of $86.60_{\pm0.34}$, while testing against the MNIST label only gives $11.61_{\pm0.30}$. We find that this also holds true for the other MSDAs. Thus, performing these distortions on the simpler data set increases its complexity to the point where it surpasses that of CIFAR-10.

Previously, we presented evidence that masking MSDA does not necessarily promote learning neither more shape nor texture information. In the light of this fact along with the results from this section, we believe image distortions force the model to learn more complex both shape and texture-specific features. Thus, in this paper we pointed out that the shift in learnt representations can lead to better models and simply quantifying the distribution shift can be misleading. An open question remains: How can we better capture the bias that is introduced and its quality? We believe understanding how a relatively small change in the data distribution impacts learnt representations could lead the way to characterising the relationship between data and model generalisation.

## 5 Conclusions

Distorting data is such a commonplace procedure, yet little effort has been devoted to investigating its broader effects. This is particularly problematic when image modifications are applied in analyses. We show a number of cases in which this leads to *biased results*. For occlusion robustness measurement, we propose an alternative. The insights we gain from this endeavour point towards the study of data characteristics as a cornerstone of our understanding and raise a number of questions about mixed sample data augmentation, on which we subsequently focus. We note that they interfere with features that are consistently found across a number of data sets and conclude that the methods commonly used are forms of mixed sample *regularisation* rather than augmentation. A limitation of previous studies that aim to explain their success is the focus on trying to argue similarity with original data, rather than explaining the bias introduced by the distortion. Correctly interpreting it is important not only for making the models trustable but also for injecting more informed prior knowledge in future applications. Beyond their practical benefits, we believe MSDAs have the potential to help characterise the interplay between data and learnt representations. Overall, the purpose of our paper is to encourage better practice when dealing with all forms of data distortions.

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
