# OpenReview forum: "On the Effects of Data Distortion on Model Analysis and Training"
_NeurIPS.cc/2021/Conference — NeurIPS 2021 Submitted_

### Official Review · Reviewer_iusD · 2021-07-15

**Rating:** 6
**Confidence:** 3

**Summary:**

In this paper, the authors examine the effects of distorting data and its effects on  biased results. They emphasize on image modifications, and they propose an alternative measure of occlusion robustness (called the interplay occlusion based on the interplay between seen and unseen data.) The authors also focus on questions about mixed sample data augmentation. They mention the need for arguing about bias from distortion and its role in trusting models and injecting informed knowledge as well.

**Limitations And Societal Impact:**

The authors mention the limitations of their proposed approach and new measure (interplay occlusion). But a separate paragraph on this topic could improve readability.
However, the societal impact still needs more details. As the real world applications and how the side effects of such bias would be  harmful need more detailed explanations.

**Main Review:**

It would have been helpful to discuss more applications in real world that would suffer from such distortion bias. Also, in the end the authors ask for better practice when dealing with all forms of data, so it could help discuss the side-effects of  this kind of bias when we have different kinds of datasets compared to the image datasets the authors explore. Some of the main experiments are mentioned (and referred to a lot) in the Appendix. So although the idea is new and the goal is clear, the writing could still improve. The significance would be more pronounced if the applications/real-world ruinous after-effects are mentioned in a separate paragraph or section.

-It is mentioned that occlusion in real-life scenarios could be caused by non-uniformly colored objects, so a suitable measure should generalize to all color patterns. It would be helpful to elaborate how this was actually done.

-It is mentioned that mostly simple features (similar to the common features) are impacted. But if the information happens to interfere with a feature that is important for separating, the performance could degrade. It could be useful to add the reasons as to why the experiment is not sufficient for general conclusions about this point and how one might try examine this issue.

-Is there a way to see what proportion of the features are the ones in the simplicity bias? That is, what proportion of them are similar to the common features.


**Time Spent Reviewing:**

3

---

> ### Author Response · Authors · 2021-08-09
> **Response to Reviewer iusD**
>
> We would like to thank the reviewer for their time and for the great suggestion to emphasize the impact our work has on real world applications. We believe this underlines the importance of our work outside its intended scope and could make our paper attractive for a wider audience.
>
> > _It would have been helpful to discuss more applications in real world that would suffer from such distortion bias. (...)  The significance would be more pronounced if the applications/real-world ruinous after-effects are mentioned in a separate paragraph or section._
>
> The most direct impact on real-world applications is in the case of model evaluation. Incorrect assessments will lead to the deployment of models which are falsely believed to be more robust than they are in reality or to models being erroneously discarded in the development phase. While the latter is less harmful, evaluating the robustness of a self-driving car using an unreliable metric can have serious implications.
>
> Additionally, as discussed in the paper, mixed-data augmentation is gaining increasing popularity. Correctly understanding the impact of these distortions is essential for trusting their usage in sensitive applications where the data can be out of distribution.
>
> We have now included these examples in the Motivation section.
>
> > _Also, in the end the authors ask for better practice when dealing with all forms of data, so it could help discuss the side-effects of this kind of bias when we have different kinds of datasets compared to the image datasets the authors explore._
>
> We would kindly like to point out that the meaning of our statement is slightly altered by the omission of a key word. In the paper we state that our manuscript encourages ''better practice when dealing with all forms of data __distortions__'' (L340--341). This is to mean both when distortions are used during model evaluation or training. Nonetheless, we agree with the reviewer that studying the impact of these distortions on other types of data such as audio or text could be an interesting future direction. The evaluation methods and some of the augmentations we study are inherently specific to image data (e.g. shape-texture bias, CutMix). Thus, we cannot directly generalise the distortions we study to other types of data.
>
> > _It is mentioned that occlusion in real-life scenarios could be caused by non-uniformly colored objects, so a suitable measure should generalize to all color patterns. It would be helpful to elaborate how this was actually done._
>
> We would like to make a slight amendment, which might have lead to the reviewer's confusion. We did __not__ claim generalisation to __all__ patterns (L187-189). We were instead interested in verifying if the two metrics are sensitive to the pattern of the occluder, thus employing random patches. The answer to this is given in the sentence that follows our statement about generalisation across colour patterns. It reads "When computing iOcclusion and CutOcclusion, we superimpose patches from images belonging to a different data set and compare the results to those obtained when occluding with black patches only" (L189--191). Thus, we occlude with patches randomly selected from other images and compare the results with those obtained when occluding with simple black patches.
>
> > _It is mentioned that mostly simple features (similar to the common features) are impacted. But if the information happens to interfere with a feature that is important for separating, the performance could degrade. It could be useful to add the reasons as to why the experiment is not sufficient for general conclusions about this point and how one might try examine this issue._
>
> We believe the reviewer refers to the experiment described in L273-380 where we perform MSDA on combined data sets. If that is correct, then this experiment was constructed as a counter-example to the claim that the augmentation quality is given by the distance from the original distribution. While the experiment is sufficient to disprove the above claim, drawing more conclusions from it about particularities of this phenomenon would be unfounded. We do not want to generalise on other aspects of this experiment since that would require finding more such cases and analysing them. We only intend to say this sort of cases exist, which contradicts the statement in question.
>
> > _Is there a way to see what proportion of the features are the ones in the simplicity bias? That is, what proportion of them are similar to the common features._
>
> To the best of our knowledge, such a tool does not exist. However, we are not sure we correctly understood the question and would politely ask the reviewer to elaborate if our answer does not correspond with the intent.
>
> Simplicity bias is a phenomenon associated with gradient-based methods [e.g 2]. The common features, including the ones we identified, are likely already subject to the simplicity bias. If the question is "what proportion of the total number of possible features is found using gradient-based methods, knowing that simplicity bias is observed?", then we are not aware of any piece of work that has answered this question.
>
> > _The authors mention the limitations of their proposed approach and new measure (interplay occlusion). But a separate paragraph on this topic could improve readability. However, the societal impact still needs more details. As the real world applications and how the side effects of such bias would be harmful need more detailed explanations._
>
> In our view, both the theoretical and the practical part of our paper could have a wider societal impact. We will focus here on the impact of our interplay occlusion measure, as requested by the reviewer.
>
> Firstly, failure to correctly choose a robust model can have serious implications in sensitive applications, as mentioned in the paper. Once again, in the case of autonomous vehicles for example, robustness to occlusion is highly relevant for the safety of all traffic participants. On the other hand, better occlusion robustness evaluation can be used in the improvement of weaponry, surveillance or other applications with potential non-humanitarian purposes.
>
> Apart from this evident category of impacts there is a more subtle one that regards reported overconfidence in models, which can undermine the expertise of medical practitioners [e.g. 1].
>
> However, this level of detail when discussing societal impact is admittedly out of our area of expertise and we can only provide a limited perspective on these complex and important problems.
>
> We politely ask the reviewer to let us know if they feel we did not fully address their concerns and we once again thank them for their time and insights.
>
> [1] Arnold, M.H., 2021. Teasing out artificial intelligence in medicine: An ethical critique of artificial intelligence and machine learning in medicine. Journal of Bioethical Inquiry, 18(1), pp.121-139.
> [2] Kalimeris, D., Kaplun, G., Nakkiran, P., Edelman, B., Yang, T., Barak, B. and Zhang, H., 2019. SGD on neural networks learns functions of increasing complexity. Advances in Neural Information Processing Systems, 32, pp.3496-3506.

---

### Official Review · Reviewer_RRFF · 2021-07-16

**Rating:** 3
**Confidence:** 1

**Summary:**

This paper studies impacts of data augmentation. Authors propose new measures for evaluating data interference and occlusion robustness. Through analyzing changes in accuracy induced by data distortion, authors argue that current practice about using and treating data augmentation has led to biased model analysis and interpretation methods.

**Limitations And Societal Impact:**

The authors adequately addressed limitations and societal impact in the paper.


**Main Review:**

The authors deal with an important area of research that discovers hidden biases in data augmentation. However, unfortunately, this paper is hard to follow. Vague words uncommon in the community are given without context or explanation. For instance, in L 43-44, what does mean by “artefacts introduced by changes in the data are negligible when evaluating models”? Similarly, what are “the artefacts introduced when training”?

Also, key measures proposed in the paper require more explanations and motivations. For instance, for data interference (Eq. 1), why is the percentage represented by the dominant class important? Similarly, why is this percentage required to be weighted by the maximum increase? In addition, it is unclear why the authors consider only increases in the error rates when we are evaluating the impacts of “data interference.” Similar issues exist for the interplay occlusion (Eq. 2).


**Time Spent Reviewing:**

15

---

> ### Author Response · Authors · 2021-08-09
> **Response to Reviewer RRFF**
>
>
> We would like to start by thanking the reviewer for the time they invested in reviewing our paper. We politely believe all of the reviewer's questions and comments find their answer in our paper. We reference the appropriate part of the manuscript and provide additional explanations. We kindly ask the reviewer to let us know if they feel anything is still unclear.
>
> > _The authors deal with an important area of research that discovers hidden biases in data augmentation. However, unfortunately, this paper is hard to follow. Vague words uncommon in the community are given without context or explanation. For instance, in L 43-44, what does mean by “artefacts introduced by changes in the data are negligible when evaluating models”?_
>
> Since this question is concerned with the meaning of a phrase rather than a specific word, we will aim to clarify the terms it contains as used in the paper, explain the context of this line and then reiterate its meaning.
>
> * distortions/changes in the data: manipulations of the original data (images) either at train or evaluation time. Visual examples of such distorted images are given in Figure 1 of the paper.
>
> * artefacts: artificial information introduced when distorting the data; spurious 'features' that result from the distortions applied to the data (e.g. the strong edges or structured noise).
>
> * negligible effect (of a phenomenon at evaluation time): the result of the evaluation is the same whether or not the phenomenon occurs.
>
> Plainly, there are evaluation methods that distort the data. This distortion has some side-effects. Until this study, it has been assumed that the side-effects do not change the result of the evaluation.
>
> More specifically, as mentioned in the Motivation section, in this paper we study two model evaluation criteria along with common and erroneous ways of measuring them: Shape-texture bias (measured by taking the accuracy when patch-shuffling images) and occlusion robustness (measured by taking the raw accuracy after superimposing patches on images, dubbed CutOcclusion). In L24--25 and L28--30 we state that the issue with patch-shuffling and CutOcclusion respectively is that they introduce information which could be misleading (i.e. artefacts). It is usually assumed this introduced information does not change the results of these evaluations (i.e. the presence of the artefacts is negligible). In other words, when evaluating models by patch-shuffling or CutOcclusion, it is assumed that the artefacts are negligible.
>
> We have now made this clearer by extending the Motivation section. We give examples of possible artefacts and we detailed the assumptions regarding the introduced information, reinforcing the connection with the line cited by the reviewer.
>
> > _Similarly, what are “the artefacts introduced when training”?_
>
> Once again, the artefacts are represented by the artificial information introduced when purposefully distorting the data.
>
> This time, we are concerned with changing the data at train time. In the paper we look at changing the data by performing mixed sample augmentation when training models. Thus, we are interested in the side-effects of performing mixed sample data augmentation. We refer to the artificial information introduced as a side-effect of mixed data augmentation as "the artefacts introduced when training".
>
>
> > _Also, key measures proposed in the paper require more explanations and motivations. For instance, for data interference (Eq. 1), why is the percentage represented by the dominant class important? Similarly, why is this percentage required to be weighted by the maximum increase?_
>
> We want to answer the question "Are artefacts negligible when analysing classifiers?". In the paper we show that artefacts are __not__ negligible when evaluating models, providing biased results. This is because the distortion can overlap with a model's learned representations of a certain class.
>
> When a distortion creates information that overlaps with a model's learned representations of a certain class, the model will tend to wrongly classify images as belonging to that class. For this reason, we look at the dominant class, as explained in L61--62 and L66--68.
>
> Let us take an oversimplified example. We want to know if measuring the superposition of a black patch creates information that tends to be associated with a certain class. That means that we will see an increase in incorrect predictions for one of the predicted classes. We look at this quantity for differently trained models. Let us assume two of the models have 100% increase in the 'Truck' class mispredictions. That is, all of the images that the models correctly classified originally and incorrectly classified when superimposing a black patch were labeled as 'Truck'. However, how can we differentiate between a model that only had an increase of 1 in the incorrect predictions and one that had 500 new incorrect predictions if we only take into account the percentage of increase? For this reason, one must weight by the increase of the highest class. We hope this toy example made the definition of our measure more intuitive.
>
>
>
>
>
> > _Similar issues exist for the interplay occlusion (Eq. 2)._
>
>
>
> It is unclear to us what part of Eq. 2 is questioned by the reviewer. For this reason, we will reiterate the whole explanation for the iOcclusion quantity. As we show in the paper, simply measuring the accuracy on unseen occluded images has two important limitations: 1. Two models with the same sensitivity to occlusion can falsely appear to have different robustness levels (L207--216); 2. The results of this evaluation are highly dependent on the specifics of the occluding patch, providing inconsistent results (L187--194, 195--206). Thus, it is insufficient to measure the accuracy on distorted test images. To address the latter limitation, we look at the gap between the train and test accuracies on occluded images. The intuition for doing so is stated in the paper in L168--172. In a massively oversimplified way, one can think of this as subtracting away the effect of the artefacts on the train data from the test data, thus cancelling it out. The denominator acts as a normalisation term, as explained in L172--173. This deals with the first limitation we mentioned.
>
> We have tried to address all the reviewer's mentioned concerns. If any part of our response is unsatisfactory, we would politely ask the reviewer to specifically point out anything that is still unclear and we will try to provide a more appropriate answer.

---

### Official Review · Reviewer_kAg1 · 2021-07-22

**Rating:** 6
**Confidence:** 3

**Summary:**

This paper delves into the assumption that data modification is detrimental to training, while negligible when analysing models. It aims to encourage better practice when dealing with data distortions rather than elimination. With experiments and analyses, the authors show that current shape biased identification methods and occlusion robustness measures are biased. The authors then propose a fairer alternative to measure occlusion robustness. A series of experiments are conducted to disprove common assumptions and put forward the argument that not preserving the data distribution can lead to learning better representations.

**Limitations And Societal Impact:**

1. As the authors mentioned in the paper, there still remains a question: 	How can we better capture the bias that is introduced and its quality.

2. It is meaningful that this work corrects and strengthen the community’s perception of how distorting data affects learning. It argues that the impacts should be understood and exploited, however, the future applications of this work are not fully investigated.

3. Some conclusion is drawn with observation rather than strict proof, which is not convincing enough, e.g. shape-biased model and texture-biased model.

4. The paper lacks an easy-to-read summary for each experiment. It is organized by posing and answering questions creatively, but it lacks a brief summary of each answer. When reading the paper, I could only understand the answers to the questions after I have finished read the whole texts. Maybe a brief and clear answer to the question should be stated ahead of the experiment and analysis part.

5. There are some minor typos in the paper, e.g. ‘asses’ in line 156 should be ‘assess’



**Main Review:**

1. The authors question the common assumption that data modality is negligible when evaluating models and important when training, and construct counter-examples against the common belief. The idea is novel and challenges the common assumption.

2. The paper does not follow the common structure of ‘Introduction - Related Work - Methodology - Experiments - Conclusion’. It is organized by posing and answering questions, which would strongly disprove common beliefs and put forward the authors’ arguments.

3. The clarity of this paper is good in general. However, the paper lacks an easy-to-read summary for each experiment. Maybe a brief and clear answer to the question should be stated ahead of the experiment and analysis part.

4. The paper aims to encourage better practice when dealing with data distortions rather than elimination. It proposes a fairer alternative named ‘interplay occlusion’ to measure occlusion robustness. It points out the limitation of previous studies and argues that correctly interpreting the effect of data distortion is of critical significance not only to make the models more trustable but also to inject more informed prior knowledge in future applications.


**Time Spent Reviewing:**

3

---

> ### Author Response · Authors · 2021-08-09
> **Response to Reviewer kAg1**
>
> We thank the reviewer for the time they devoted to this paper. We believe the suggestion to provide a summary for each experiment made our paper easier to follow. Unfortunately, it was unclear to us from the review what other improvements we can bring to the paper or clarifications we could give that would persuade the reviewer to change their score.
>
>
>
> > _The clarity of this paper is good in general. However, the paper lacks an easy-to-read summary for each experiment. Maybe a brief and clear answer to the question should be stated ahead of the experiment and analysis part._
>
>
>
> This is a great suggestion! We now state the answer to the question right after posing it and we believe this improves understandability.
>
>
>
> > _As the authors mentioned in the paper, there still remains a question: How can we better capture the bias that is introduced and its quality._
>
> We regard this open question as future work rather than a limitation. This question is complex and we believe deserves a paper of its own. We believe it is intimately linked to how one can describe the desirable qualities of a data distribution and with the relationship between data and learned representations, both of which are important and not trivial to answer. As the reviewer has noted, the focus of the current manuscript is to challenge common assumptions and does not aim it fully describe the implications of data distortion. We believe questioning these wrong assumptions is critical and that it is important for researchers in the field understand the pitfalls of following an incorrect viewpoint. We see this as the key contribution of our paper.
>
>
>
> > _It is meaningful that this work corrects and strengthen the community’s perception of how distorting data affects learning. It argues that the impacts should be understood and exploited, however, the future applications of this work are not fully investigated._
>
> Although our work can be used for practical application, we believe the most important contribution is in combating erroneous research directions and improving understanding. On the practical side, the occlusion robustness measure can represent a new standard for assessing sensitivity to occlusion, which can lead to the development of stronger models. Similarly, our DI measurement can be used to identify more cases of data interference, hence leading to more principled practices when dealing with data. But most importantly, we hope our work sets a new direction in understanding how distorting training data impacts learning. As mentioned above, we believe this could be used in better understanding the data and learned representations, which could ultimately play a small role in understanding generalisation.
>
>
>
> > _Some conclusion is drawn with observation rather than strict proof, which is not convincing enough, e.g. shape-biased model and texture-biased model._
>
>
>
> All our proofs are proofs by counterexample often used in mathematics, philosophy and logic. Unfortunately, we are not aware of the existence of any rigorous mathematical framework for any of the problems we study that would allow us to provide an alternative proof. Could the reviewer be more precise as to which conclusion is not convincing enough and we could perhaps make the logical argument clearer?
>
>
>
> > _There are some minor typos in the paper, e.g. ‘asses’ in line 156 should be ‘assess’._
>
>
>
> We thank the reviewer very much for pointing out this typo!
>
> We are happy to answer any other questions or provide additional explanations.

---

> > ### Comment · Reviewer_kAg1 · 2021-08-24
> > **I keep my score after reading the author response.**
> >
> > Thank you for your comment. After reading other reviews and going through the paper a little bit, I vote for accept (6, borderline accept), but I will not be disappointed if this paper is finally rejected.
> >
> > This is an analysis paper on data augmentation, the resulting models, bias of previous methods, etc. To be honest, I'm not an expert in this narrow area. To me, it is an interesting paper, but not that satisfying.
> >
> > It is interesting to know some drawbacks of existing techniques and evaluation protocols. However, it still lacks insights of the data bias in learning procedure. It is not necessarily theoretical proof, but insights / intuition / explanation could also be helpful. As a result, it seems not easy to know the applications (e.g., how to improve the models) after reading the papers. I absolutely understand it is not easy to make everything in a single paper, but it is what a scientific paper needs to do, isn't it?
> >
> > As a summary, it is good, but not exciting, thus I keep my rating. Finally, I'm happy to find the authors will a summary before each analysis section. This is particularly important for the readability as an analysis paper.

---

> > > ### Author Response · Authors · 2021-08-25
> > > **Unclear criticism**
> > >
> > > We thank the reviewer for their response, however, with all due respect, we do not believe the purpose of a scientific paper is to satisfy and excite, nor should it be an evaluation criteria. The quality of a piece of work is usually judged by the novelty and importance of its contributions. After carefully researching and understanding the issues in this area, we strongly believe our contributions are important. Despite the field's focus on eye-catching results, we argue advancements in understanding are equally important and should not be dismissed. Further we would highlight that without concrete work on understanding, more and more ("exciting") papers will continue to be published based on highly flawed assumptions and evaluations.
> > >
> > > We respectfully disagree that "everything needs to be made in a single paper", with no future work left to be done. In the history of scientific publishing we would argue that the majority of papers do not answer all questions they raise. Progress in science has generally resulted from accumulated evidence over many publications. As an example, the Mixup paper [3] does not answer the most immediate question: "Why does Mixup work?". Instead, a number of __separate__ papers were written to tackle this [e.g. 1, 4].
> > > A paper that is closer to our approach of combatting existing research direction is _Zhang et al. (2016)_ [2]. They empirically show deep models have such vast capacities that statistical learning bounds become vacuous. They do not, however, answer the question of what a good framework for reasoning about generalisation is.
> > > These are two straightforward examples. If the reviewer believes more are needed to support our claim, then please let us know and we will bring more supporting evidence.
> > >
> > > As with our previous response, we would kindly ask the reviewer to specifically mention what part of the paper needs more "insights / intuition / explanation" for it to be "convincing enough". For each question we raise and experiment we perform we provide insights and explanations and where appropriate, the intuition. Thus, it is unclear to us which specific part the reviewer refers to. We also highlight that the entirety of Section 3 is about how to practically improve models, as it provides a set of concrete measures that are demonstrably fairer for evaluating model performance. Other researchers could immediately adopt these measures in order to advance the state of the art in real-world machine learning problems.
> > >
> > > [1] Carratino, L., Cissé, M., Jenatton, R. and Vert, J.P., 2020. On mixup regularization. arXiv preprint arXiv:2006.06049
> > >
> > > [2] Zhang, C., Bengio, S., Hardt, M., Recht, B. and Vinyals, O., 2021. Understanding deep learning (still) requires rethinking generalization. Communications of the ACM, 64(3), pp.107-115.
> > >
> > > [3] Zhang, H., Cisse, M., Dauphin, Y.N. and Lopez-Paz, D., 2017. mixup: Beyond empirical risk minimization. International Conference on Learning Representations, 2018.
> > >
> > > [4] Zhang, L., Deng, Z., Kawaguchi, K., Ghorbani, A. and Zou, J., 2020. How Does Mixup Help With Robustness and Generalization?. International Conference on Learning Representations, 2021.

---

### Official Review · Reviewer_VZ6B · 2021-08-02

**Rating:** 7
**Confidence:** 3

**Summary:**

This paper studies empirically the effect of artefacts and biases in model analysis and training methods relying on data manipulation. They argue that common metrics used for analysing shape bias and occlusion robustness also (undesirably) include the effect of artefacts (such as edge sharpness) introduced by the corresponding image manipulations, and propose a new metric, iOcclusion, which is more robust to these effects.  Further, it is argued that these artefacts also manifest themselves in training schemes based on mixed sample data augmentation (MSDA), and can be beneficial by introducing a systematic bias towards complexity.

**Limitations And Societal Impact:**

Yes, limitations and societal impact are adequately discussed.

**Main Review:**

Overall, this paper makes a number of interesting and novel insights regarding current practices in data modification for image classification. These are well supported for the most part by (clear and well-designed) experimental evaluations, though the lack of formal definitions and analysis means that some claims remain somewhat speculative. There are some issues with the clarity of the narrative, though I believe this is non-critical and can be fixed.

The paper firstly introduces the DI metric and show that models can have similar shape biases (according to GST dataset), but perform differently under DI w/ shuffling. It is then argued that this shows measures based on patch-shuffling are unreliable. However, since DI was introduced by the authors, I think a better representative for “effect of patch-shuffling” would be accuracy after shuffling, and how this compares to GST. At least, it could be discussed how the class-focused DI compares to accuracy. **AR**: I am satisfied with the author response and it is good to see that there is still a discrepancy between accuracy after shuffling and GST.

The authors then make a similar argument for occlusion robustness. However, it is not clear to me what "occlusion robustness" actually means, since there is no formal definition, nor clear empirical gold standard like GST dataset for shape bias. In particular, since the model is treated as a black-box for all occlusion robustness metrics (including iOcclusion), how can one distinguish whether the model’s representation is “not occlusion robust” or “using edge bias/artefacts”, if this leads to the same input-output behaviour, i.e. misclassification? **AR**: Thank you for the explanation. If I understand correctly, the distinction is between whether the performance only drops for some types of occlusion such as a box (indicating sensitivity to artefacts), or all types of occlusion (indicating lack of robustness/inability to classify without the occluded information). Firstly, I would suggest the authors define this formally as the target/proxy to the intuitive notion of "how much information can be hidden from the network". Secondly, I accept the authors' point that iOcclusion has been shown to be less sensitive to extraneous factors such as edge artefacts. However, invariance is only a necessary, and not sufficient condition to show that it captures occlusion robustness. For instance, a function which randomly assigns a value to each network would also be invariant. While Figure 3 does indeed show that FMix is given a greater iOcclusion robustness than the basic network, more analysis is needed to definitively prove this link/correlation between occlusion robustness and iOcclusion. For instance, one could assess, for a fixed occlusion type, how correlated *accuracy drop* is with *iOcclusion*, with respect to different networks.

The definition of iOcclusion using the train-test generalization gap to improve consistency (w.r.t. colour/shape pattern of occlusion patch) of the metric is a very interesting and, to the best of my knowledge, novel idea. I find the argument regarding consistency with weaker models (Lines 207-216, Figure 4) comparatively weaker as I expect CutOcclusion to also have this property if it was instead defined as the drop in accuracy, as is commonly done.

Finally, the authors argue against existing model-based measures of data augmentation quality such as diversity and affinity, and suggest to instead focus on introduced biases. This is well supported by the inter-dataset experiments.

In terms of clarity, I found the paper very readable and well-written on the small scales, but I felt the narrative of the paper is sometimes muddled and unclear, and the "big picture" can get lost in the minutiae. For instance, a clear statement of the desiderata of iOcclusion at the start of Section 3 would be helpful. Also, I found the Introduction to be quite difficult to follow without having read the rest of the paper beforehand. For instance, I found the statement in Lines 43-44 “In summary, it is currently assumed that the artefacts introduced by changes in the data are negligible when evaluating models, while those introduced when training are important and undesirable.” to be very difficult to interpret without having read the whole paper to understand what is meant by “negligible” and “important and undesirable”. I think the paper would be improved if the Introduction focused more on examples and the intuitive argument. **AR**: The authors have agreed to address clarity issues (also raised by other reviewers); I would once again strongly encourage the authors to carefully reassess each section to make sure a) the idea is clear without reading the whole section and b) all quantities and concepts  are fully defined and the logical argument is made as clear as possible. Clarity is always important but especially paramount for papers such as this which follow an unconventional structure and whose main contribution is a critical analysis of previous work.

**AR**: In light of the authors' response, I have increased my score and vote to accept the paper. As the other reviewers have pointed out and the authors have noted, the study mostly focuses on negative results showing counterexamples to prior work rather than new solutions. However, in my opinion this is not a key limitation as such results are also important, and the paper makes a number of interesting observations, such as the bias of artefacts in robustness evaluation metric, backed up with empirical evidence. If the logical arguments are laid out more clearly, along with the adjustments that the authors have agreed to make, I believe that the paper consititutes a good contribution to the area which will be of utility to other researchers.

Other questions:

• In Figure 3, for fraction removed 0.1 and the FMix model, iOcclusion is much greater than one, meaning the generalization gap is larger for occluded images than for the original images. Do the authors have an explanation for why this is?

• In the definition of DI, is increase in misclassification for class $c$ calculated for images with true label $c$, or predicted as $c$ by the classifier?



**Time Spent Reviewing:**

8

---

> ### Author Response · Authors · 2021-08-09
> **Response to Reviewer VZ6B**
>
> We would first like to genuinely thank the reviewer for their time, useful suggestions and thoughtful questions. We really feel the valuable feedback has strengthened our paper.
>
> In our response we specifically ask the reviewer about the questions we are unsure we correctly understood. However, it is possible that we misinterpreted other comments or questions, in which case we would cordially ask the reviewer to point this out to us and we will try to reply as promptly as possible.
>
> > _Overall, this paper makes a number of interesting and novel insights regarding current practices in data modification for image classification. (...) There are some issues with the clarity of the narrative, though I believe this is non-critical and can be fixed._
>
> We are happy to take suggestions on the narrative, in addition to the useful ones already mentioned in the latter part of the review, and will make changes accordingly.
>
> > _The paper firstly introduces the DI metric and show that models can have similar shape biases (according to GST dataset), but perform differently under DI w/ shuffling. It is then argued that this shows measures based on patch-shuffling are unreliable. However, since DI was introduced by the authors, I think a better representative for “effect of patch-shuffling” would be accuracy after shuffling, and how this compares to GST. At least, it could be discussed how the class-focused DI compares to accuracy._
>
> Unfortunately, the DI measure cannot be compared to accuracy after shuffling because they measure different things. The accuracy measures the overall effect of patch-shuffling. This is used in some studies to measure the texture bias of models [e.g. 2, 4], but we argue there are side-effects of patch-shuffling that make it unreliable to measure texture bias. The DI is concerned with the __side-effects__, not with the texture bias itself. The DI measure allows us to identify the extent to which a model associates patch-shuffled images with a particular class because of the artificial information introduced.
>
> We use the GST experiment to show that the side effects that we observe are not necessarily associated with higher shape bias. This is crucial for our argument against purely measuring accuracy after patch shuffling. It shows that two models with the __same shape bias__ can have __different accuracies__ when patch-shuffling if for one of them the learned representations happen to overlap with the artefacts introduced by shuffling. The overlap is the side-effect our DI measure captures.
>
> However, we thank the reviewer for this comment. We have now included in the paper the accuracy results along with an explanation of how they differ from the DI measurement. We believe this makes our argument even clearer. The additional experiment shows there exist inconsistencies between the results of GST and accuracy which the DI can explain.
>
> > _The authors then make a similar argument for occlusion robustness. However, it is not clear to me what "occlusion robustness" actually means, since there is no formal definition, nor clear empirical gold standard like GST dataset for shape bias._
>
> This is a fair point and we will clarify this now and in the paper: Occlusion robustness should reflect how much information about the object can be hidden from the model without affecting the model's ability to classify, which is the common take on occlusion robustness [e.g. 1,3]. Thus, it is concerned with the __absence__ of features, not with the presence of occluding features.
>
> One can think of an oversimplified example where dogs are correctly classified when the face can be seen and incorrectly classified when the face cannot be seen. In such a case, we would consider the model is not robust to occlusion.
>
> This is to be distinguished from a model that would normally correctly classify dogs when the face is hidden. However, if a rectangular patch is artificially introduced and the model associates that feature with a different class (e.g. Truck), in this case the misclassification is caused by the artefacts and not by the missing feature.
>
>
> > _In particular, since the model is treated as a black-box for all occlusion robustness metrics (including iOcclusion), how can one distinguish whether the model’s representation is “not occlusion robust” or “using edge bias/artefacts”, if this leads to the same input-output behaviour, i.e. misclassification?_
>
> We do agree with the reviewer that both removing relevant features and introducing artificial information can have the same impact on classification. This is precisely the problem we have identified. To reiterate, a practitioner evaluating the model with the typical (CutOcclusion) measure would only be able to tell that the model is misclassifying the distorted images. As the reviewer mentioned, they would not be able to distinguish between the misclassification being caused by the absence of features or the presence of artefacts. However, this distinction in what is causing the model to misclassify is important for model analysis and improvement, and this is exactly what we are trying to capture with the iOcclusion measure. By its definition, our measure is aiming to lessen the impact of the artefacts (caused by e.g shape or colour of the occluding patch) through the train-test gap, precisely as the reviewer has noted in the following paragraph of the review.
>
> We are not entirely sure we have correctly understood the reviewer's question. As we would like to fully address any concerns, we would kindly ask the reviewer to further clarify anything that we did not properly answer and we will do our best to give a complete answer.
>
> > _I find the argument regarding consistency with weaker models (Lines 207-216, Figure 4) comparatively weaker as I expect CutOcclusion to also have this property if it was instead defined as the drop in accuracy, as is commonly done._
>
> We respectfully disagree that the occlusion robustness is commonly defined as the drop in accuracy (by which we assume the reviewer means the difference between the accuracy on clean and occluded data). In all 5 papers that we cite for CutOcclusion, the robustness is measured as the accuracy on occluded images, which is what Figure 4 (left) depicts in the paper. We are unaware of any recent studies that measure it as the drop in accuracy.
>
> We agree that if this was be the common way of assessing robustness, our experiment would be redundant. However, all the studies we have read follow the procedure we describe in the paper. We might be unaware of papers that assess the difference in accuracy. Even in this case, given the increased number of recent papers that simply measure the accuracy, we believe it is important to point this issue out through the comparison we have made. Given the above, we would kindly ask the reviewer to let us know if they still find the comparison weak.
>
> > _For instance, a clear statement of the desiderata of iOcclusion at the start of Section 3 would be helpful._
>
> This is an excellent idea. We agree that it will help the reader understand the bigger picture before delving into the details.
>
> > _Also, I found the Introduction to be quite difficult to follow without having read the rest of the paper beforehand. For instance, I found the statement in Lines 43-44 “In summary, it is currently assumed that the artefacts introduced by changes in the data are negligible when evaluating models, while those introduced when training are important and undesirable.” to be very difficult to interpret without having read the whole paper to understand what is meant by “negligible” and “important and undesirable”. I think the paper would be improved if the Introduction focused more on examples and the intuitive argument._
>
> Thank you for your detailed suggestions! We expanded the Motivation section and focused on clarity for first-time readers. Particularly, we detailed the assumptions regarding artefacts right when they were first mentioned for each type of distortion (L24--25, 28--30, 41--42) and reinforced the connection with the summary mentioned by the reviewer.
>
> [1] Rajaei, K., Mohsenzadeh, Y., Ebrahimpour, R. and Khaligh-Razavi, S.M., 2019. Beyond core object recognition: Recurrent processes account for object recognition under occlusion. PLoS computational biology, 15(5), p.e1007001.
>
> [2] Shi, B., Zhang, D., Dai, Q., Zhu, Z., Mu, Y. and Wang, J., 2020, November. Informative dropout for robust representation learning: A shape-bias perspective. In International Conference on Machine Learning (pp. 8828-8839). PMLR.
>
> [3] Tang, H., Schrimpf, M., Lotter, W., Moerman, C., Paredes, A., Caro, J.O., Hardesty, W., Cox, D. and Kreiman, G., 2018. Recurrent computations for visual pattern completion. Proceedings of the National Academy of Sciences, 115(35), pp.8835-8840.
>
> [4] Zhang, T. and Zhu, Z., 2019, May. Interpreting adversarially trained convolutional neural networks. In International Conference on Machine Learning (pp. 7502-7511). PMLR.

---

> > ### Author Response · Authors · 2021-08-09
> > **Response to Reviewer VZ6B**
> >
> > > _Other questions: In Figure 3, for fraction removed 0.1 and the FMix model, iOcclusion is much greater than one, meaning the generalization gap is larger for occluded images than for the original images. Do the authors have an explanation for why this is?_
> >
> > For small occluding patch sizes, models trained with masking augmentations have a small empirical error, which causes the occlusion gap to be bigger. This is expected, given the intuition behind the proposed measure. For more robust models, the artefacts will not have such a high impact on training data as they do on test, thus leading to an enlarged gap. As the size of the patch increases and the robustness decreases, both the train and test data will be equally affected and the gap will fall off to 0. Additionally, masking methods generally have a small generalisation gap, making this phenomenon a common one in their case.
> >
> > > _In the definition of DI, is increase in misclassification for class 'c' calculated for images with true label , or predicted as 'c' by the classifier?_
> >
> > This is a very good question. Thank you for pointing out that this is not made clear enough in the paper. It is 'c' as predicted by the classifier, since we are interested in determining whether the distortion is consistently associated with a particular class (e.g. the images with overlapping squares are wrongly associated with class 'Truck'), rather than which true class is most affected by the distortion.
> >
> > We once again thank the reviewer for their extensive feedback and we hope we clarified any misunderstanding. We kindly wait for the reviewer's response.

---

> > ### Author Response · Authors · 2021-08-23
> > **Clarification on comparison with accuracy after shuffling**
> >
> > We would like to clarify our response to the reviewer's request to compare accuracy after shuffling with GST results and to DI.
> >
> > In our initial response we have possibly not made it clear enough that we computed the accuracy after shuffling and it is indeed giving different results to the GST approach, as we expected. Thus, to the reviewer's observation that accuracy after shuffling should be compared to GST, we now explicitly include this comparison which is in line with the statements of the paper.
> >
> > We also now clarify in the paper the difference between what the DI measure captures and what accuracy captures, and explain why they cannot be directly compared.

---

> > ### Comment · Reviewer_VZ6B · 2021-08-25
> > **Update after Author Response**
> >
> > I would like to thank the authors for their comprehensive response; I have read it carefully and updated my review accordingly.

---

> > > ### Author Response · Authors · 2021-08-25
> > > **Thank you for your response!**
> > >
> > > We once again thank the reviewer for their involvement in reviewing and improving our paper, for all the positive feedback and additional suggestions.
> > >
> > > Yes, the reviewer correctly understood our explanation and we agree that the definition mentioned adds to the clarity and precision of the paper. We now include this in our manuscript. We also make the distinction drawn by the reviewer more explicit in the paper.
> > >
> > >
> > > We agree with the reviewer that invariance is necessary, but not sufficient. However, in the absence of a ground truth, the correlation between occlusion robustness and iOcclusion cannot be fully proved.
> > >
> > > We have computed the correlation requested by the reviewer and obtained an average Pearson correlation coefficient of over 0.97 for all the PreAct-ResNet models we study in the paper (basic, Mixup, FMix and CutMix models). The average is computed across 5 different model runs and we fix the occlusion type to random rectangular black patches. This result appears to be positive and was expected given that the definition of iOcclusion already accounts for the drop in accuracy.
> > >
> > > We are, however, reluctant to add this correlation result to the paper; indeed, accuracy drop is indicative up to a point, but as we argue in our work, this is not fully descriptive and _on its own_ it can fail as a proxy to true occlusion robustness. If the reviewer still deems it necessary, we can add this result to the manuscript for future reference, but with the caveat that we do not believe this adds supporting evidence for the link between true occlusion robustness and iOcclusion.

---

### Decision · Program_Chairs · 2021-09-27

**Decision:**

Reject

**Comment:**

This paper studies the interesting topic of the artefacts and biases that can be introduced by approaches that rely on data manipulation.

After the discussion period, the paper received final scores of 7663, but where the 3 was a very low confidence review with little detail and for which the reviewer did not participate in discussions or requests to add more detail to their review.  I have predominantly thus discounted this review from my deliberations, but not entirely as I do think the failure of this reviewer to understand the paper is actually partly representative of clarity issues of the work raised by other reviewers.  It is also worth noting that this was a paper where it proved extremely difficult to recruit appropriate reviewers, with very few bids received in the bidding process and numerous assigned reviewers pulling out because they did not feel they could sufficiently well understand the work to assess it; something which I think is again reflective of clarity issues as discussed below.

Overall, I see this paper as very much on the borderline, with reviewers praising some of its underlying content and insights, but repeated issues raised on clarity and reviewers sometimes unsure about the paper's contributions.  Indeed, I myself struggled substantially when I tried to read the paper and believe it currently requires substantial effort to understand.  In particular, I think the introduction (and to a lesser extent the abstract) are both very poor at outlining what the paper is trying to achieve and giving motivation and context for the work.  I think one reviewer puts it very well when they say "I found the Introduction to be quite difficult to follow without having read the rest of the paper beforehand" which is exactly the opposite of how a paper should read: the introduction is usually the most critical part of the paper and lays the foundation for the reader to understand the work.  Similarly, issues then percolate down through the paper, which, amongst other things, is regularly too vague and imprecise; often lacks a central narrative and fails to properly link the lower level ideas back into the higher level aims; and generally lacks the hand-holding and high-level glue that make a paper easy to follow.

In light of this, the keys questions become whether the strengths of the paper are sufficient for these shortfalls on clarity to be overlooked and whether the required adjustments can reasonably be expected to be made in time for the camera-ready.  Though I think this is a very tight decision (which I mulled over for some time), unfortunately, my final conclusion is that too much needs to be done for the paper to be accepted at this time (e.g. I think the introduction needs a complete rewrite), bearing in mind the highly competitive nature of this review process.  I hope though that the reviews and discussions have been helpful to the authors and that they will make the required updates and successfully resubmit to another venue.